# Design and Psychometric Evaluation of the ‘Clinical Communication Self-Efficacy Toolkit’

**DOI:** 10.3390/ijerph16224534

**Published:** 2019-11-16

**Authors:** José Manuel Hernández-Padilla, Alda Elena Cortés-Rodríguez, José Granero-Molina, Cayetano Fernández-Sola, Matías Correa-Casado, Isabel María Fernández-Medina, María Mar López-Rodríguez

**Affiliations:** 1Nursing, Physiotherapy and Medicine Department, Faculty of Health Sciences, Universidad de Almeria, 04007 Almería, Spain; j.hernandez-padilla@ual.es (J.M.H.-P.); jgranero@ual.es (J.G.-M.); cfernan@ual.es (C.F.-S.); mcc249@ual.es (M.C.-C.); isabel_medina@ual.es (I.M.F.-M.); mlr295@ual.es (M.M.L.-R.); 2Adult, Child and Midwifery Department, School of Health and Education, Middlesex University, London NW4 4BT, UK; 3Associate Researcher, Faculty of Health Sciences, Universidad Autónoma de Chile, Temuco 4780000, Chile

**Keywords:** clinical communication, psychometrics, self-efficacy

## Abstract

Nursing students experience difficulties when communicating in clinical practice. Their self-efficacy in clinical communication should be explored as part of their competence assessment before they are exposed to real human interactions in the clinical setting. The aim of this study was to design and psychometrically evaluate a toolkit to comprehensively assess nursing students’ self-efficacy in clinical communication. The study followed an observational cross-sectional design. A sample of 365 nursing students participated in the study. The ‘Clinical Communication Self-Efficacy Toolkit’ (CC-SET) was comprised of three tools: the ‘Patient-Centered Communication Self-efficacy Scale’ (PCC-SES), the ‘Patient clinical Information Exchange and interprofessional communication Self-Efficacy Scale’ (PIE-SES), and the ‘Intrapersonal communication and Self-Reflection Self-Efficacy Scale’ (ISR-SES). The tools’ reliability, validity (content, criterion, and construct) and usability were rigorously tested. The Cronbach’s alpha coefficient of the three tools comprising the CC-SET was very high and demonstrated their excellent reliability (PCC-SES = 0.93; PIE-SES = 0.87; ISR-SES = 0.86). The three tools evidenced to have excellent content validity (scales’ content validity index > 0.95) and very good criterion validity. Construct validity analysis demonstrated that the PCC-SES, PIE-SES, and ISR-SES have a clear and theoretically-congruent structure. The CC-SET is a comprehensive toolkit that allows the assessment of nursing students’ self-efficacy in interpersonal, interprofessional, and intrapersonal communication.

## 1. Introduction

All nurses are expected to be competent in clinical communication [1]. This means that they have to be able to effectively communicate with patients, patients’ relatives and colleagues, as well as with themselves as part of the self-reflection process [2,3]. Nurses’ ability to communicate effectively has been linked to positive patient outcomes [4,5,6], increased patient safety [7,8], and improved nurse-related outcomes [7,9]. In response, clinical communication education has become an integral part of undergraduate nursing programs [10]. In fact, in recent years, significant efforts have been made to design, implement, and evaluate innovative educational strategies that have aimed to foster the acquisition and retention of competence in clinical communication amongst nursing students [11,12,13]. However, evidence suggests that competence in clinical communication can be difficult to acquire and nursing students experience difficulties when communicating with patients, patients’ relatives, and other colleagues during their clinical placements [3,14,15,16]. It is, therefore, important that nursing students are educated in clinical communication before they are exposed to clinical settings [17].

Clinical communication is a wide and multidimensional concept that encompasses: (1) interpersonal communication, which refers to the process of exchanging ideas, thoughts, feelings and needs with another person [3]; (2) interprofessional communication, which refers to the process of conveying information with another colleague [18]; (3) intrapersonal communication, which refers to the internal mental processes by which individuals reflect on themselves and their communicational interactions with others [19]. In line with this, international literature shows that nursing students are often educated to conduct person-centered conversations (interpersonal communication) [11,15], to communicate patients’ clinical information using the SBAR method (interprofessional communication) [13,18,20] and to communicate with oneself or self-reflect (intrapersonal communication) [21,22]. However, it remains unclear whether nursing students’ competence in clinical communication is both acquired and comprehensively assessed before they are exposed to real human interactions during their clinical placements [17,20].

Becoming competent in clinical communication requires gaining not only theoretical knowledge and practical skills but also self-efficacy in one’s ability to implement the behavioral actions needed to attain communicational goals in the clinical setting [21]. Following Bandura’s theory, self-efficacy is the level of confidence an individual has in his own ability to accomplish a task [21,23,24]. It is believed that individuals with higher levels of self-efficacy are more likely to mobilize their motivation and cognitive resources towards implementing the actions required to achieve a goal [12,24]. Therefore, the assessment of self-efficacy should be part of the assessment of nursing students’ competence during their clinical training.

Assessing nursing students’ self-efficacy can help nursing educators to better understand the effects of their educational interventions and the real educational needs of their students [24]. In fact, self-efficacy is often considered a reliable educational outcome in clinical communication education and has been assessed in numerous studies over the past few years [12,13,21,25]. However, none of the instruments used in these studies have been specifically designed and validated to assess nursing students’ self-efficacy in their ability to conduct person-centered conversations, to communicate patients’ clinical information using the SBAR method and/or to communicate with themselves. The aim of this study is to design and psychometrically evaluate a toolkit to comprehensively assess nursing students’ self-efficacy in clinical communication.

## 2. Materials and Methods

### 2.1. Study Design and Participants

This study followed an observational cross-sectional design. Nursing students from a southern Spanish university were recruited for participation using a convenience sampling technique. The data were collected between April 2017 and February 2018. The inclusion criteria for participating in the study were: to be ≥18 years old, and to be enrolled in a Nursing degree program during the 2016/2017 academic year. Four hundred and eighty participants were eligible for inclusion and 365 volunteered to participate in the study. Participants’ demographic characteristics included: age, gender, level of education completed, and year group (program cohort). To allow for later between-groups comparisons, the time since the participant last attended a training session in clinical communication was also recorded.

### 2.2. Ethical Considerations

Ethical approval was granted by the Research Ethics Committee of the Department of Nursing, Physiotherapy and Medicine at the University of Almería (Spain) before commencing the recruitment process (EFM-10/15). A member of the research team invited the students to participate and provided them with a written document with information about the participants’ rights, the study’s aim and the data collection process. Students who volunteered to take part in the study signed an informed consent form before participating. All data collected were treated in accordance with the European legislation on data protection.

### 2.3. Initial Development and Testing of the Toolkit

The initial version of the ‘Clinical Communication Self-Efficacy Toolkit’ (CC-SET) was developed in Spanish and included 31 items comprising three tools: the ‘Person-centred Communication Self-Efficacy Scale’ (PCC-SES), the ‘Patient Clinical Information Exchange and Interprofessional Communication Self-Efficacy Scale’ (PIE-SES), and the ‘Intrapersonal Communication and Self-Reflection Self-Efficacy Scale’ (ISR-SES).

The researchers developed the CC-SET based on different communication methods and theories. The PCC-SES was comprised of 17 items and it was based on the basic principles of effective interpersonal communication and two person-centred communication models [26,27]. These two models provide a clear structure to conduct patient-centred conversations using basic communication skills to help patients and families to find their own solutions to their concerns [26,27]. On the one hand, in the SAGE & THYME**^®^** model, healthcare professionals must do the following: (S) consider the appropriateness of the Setting in which they start the conversation, (A) Ask what the person’s concerns are, (G) Gather all the person’s concerns, (E) use Empathy when responding to the other person, (T) find out whether the person has somebody to Talk to, (H) ask whether the people they can talk to could be of Help, (Y) ask the person “what do You think would help?”, (M) ask the person “is there something you would like Me to do?”, and (E) End the conversation by summarizing it [26]. On the other hand, according to the SEGUE model, when healthcare professionals communicate with patients and their relatives, they must follow these steps: (S) Set the stage, (E) Elicit information, (G) Give information, (U) Understand their perspective, and (E) End the encounter [27]. The PIE-SES was comprised of 6 items and it was based on the SBAR model to communicate patients’ clinical information to other healthcare professionals [28]. This model provides a set structure to communicate clinical information about a patient to other colleagues [28]. Following the SBAR model, healthcare professionals have to assertively state the patient’s Situation at the time of contact, the patient’s Background, the Assessment carried out so far, and some Recommendations as to how to go about possible solutions for the situation [28]. The ISR-SES was comprised of 8 items and it was based on Rogers’ and Honeycutt’s theories on how the person’s ‘self’ plays a crucial role in the human communication process [19,29]. These two theories argue that for humans to effectively reflect on their experiences and learn from them, they need to establish a skilled internal dialogue in which the person must be fully aware of who he/she is [19,29].

All the items comprising the CC-SET measured nursing students’ confidence in terms of ‘can do’ using a Likert scale with response options that range from 0–100 [23,24]. Additionally, in order to avoid a ceiling effect on the participants’ scores, a certain level of difficulty was added to all the statements comprising the three tools [23]. This was achieved by making the items as specific as possible and by adding elements that would make participants question whether they would be able to correctly perform each task in every possible situation. For example, in item 19, instead of only stating the generic action “communicate the patient background”, some elements were added in order to make the task more specific and more difficult to achieve: “communicate the patient background in a comprehensive yet concise manner without forgetting any information”.

Before piloting the CC-SET amongst nursing students, a panel of 17 independent experts in healthcare communication from 13 different institutions were recruited to critically revised the PCC-SES, PIE-SES, and ISR-SES by judging the relevance of each item and commenting on their wording. The experts had to meet the following criteria: (1) to be a qualified nurse or medical doctor with, at least, a MSc qualification; (2) to have taught communication skills for healthcare professional for more than 5 years; and (3) to have experience with the development of psychometric instruments. In order to assess the CC-SET’s content validity, the 17 experts were asked to independently complete an email survey that allowed for the calculation of the tools’ content validity index (CVI) [30]. In this regard, the experts individually scored the items as 1 = ‘not relevant’, 2 = ‘somewhat relevant’, 3 = ‘quite relevant’, or 4 = ‘highly relevant’ for measuring nursing students’ self-efficacy on their ability to effectively do the following: (1) conduct person-centered communicational interactions (PCC-SES), (2) exchange patients’ clinical information and communicate with other healthcare professionals (PIE-SES), and (3) communicate with oneself and self-reflect (ISR-SES). Each item’s CVI (I-CVI) was calculated by adding the number of experts scoring the item as either ‘quite relevant’ or ‘highly relevant’ and dividing it by the overall number of experts comprising the panel [30]. In a panel with 17 experts, an I-CVI ≥ 0.78 can be accepted as evidence of a high degree of agreement about the relevance of the item to measure the intended construct [30]. Therefore, all items whose I-CVI ≥ 0.78 were kept as part of the PCC-SES, PIE-SES, and ISR-SES.

Once the expert revised the PCC-SES, PIE-SES, and ISR-SES, the toolkit was tested amongst a pilot sample (*N* = 60). The same recruitment criteria and ethics protocol used for the main sample were also applied to the pilot sample. The 60 participants comprising the pilot sample did not participate in the main validation study. To allow for the assessment of the tools’ reliability and temporal stability, the pilot sample completed the CC-SET in two occasions separated by a 6-week interval. The temporal stability of the tools was evaluated by calculating the Pearson’s correlation coefficient (r) between the test-retest results. In this 6-week interval, participants did not receive any training in clinical communication. The tools’ reliability was explored using three estimators: (1) the Cronbach’s coefficient alpha (α) for each tool, (2) the corrected item-total correlation (C-ITC), and (3) the tool’s estimated Cronbach’s α if a particular item was removed. Items were retained as part of the tools if: (1) the item’s C-ITC > 0.3, and (2) the instrument’s Cronbach’s α did not significantly increase after removing that item.

In order to evaluate the readability and understandability of the PCC-SES, PIE-SES, and ISR-SES, the experts and the participants provided feedback on the wording of each tool and how well they understood them.

The results of the pilot study of the three tools comprising the CC-SET are presented below. 

#### 2.3.1. Person-Centered Communication Self-Efficacy Scale (PCC-SES)

The I-CVI of the 17 items comprising the PCC-SES ranged from 0.88–1 and they were all retained for its pilot study. The PCC-SES’s temporal stability (r = 0.79) and reliability (all items’ C-ITC > 0.3; Cronbach’s α = 0.91, which would not have significantly increased if any of the items had been removed) proved to be very good after its pilot study. Although most students and experts stated that the PCC-SES was easy to understand and complete, nine individuals recommended rewording some items to make them clearer and more explicit. These comments were taken into consideration and the affected items were reworded. Please see Appendix A for detailed information about the comments made by experts and participants.

#### 2.3.2. Patient Clinical Information Exchange and Interprofessional Communication Self-Efficacy Scale (PIE-SES)

The I-CVI of the 6 items comprising the PIE-SES ranged from 0.88–1. All of them were retained as part of the tool. The pilot study evidenced that the PIE-SES was temporally stable (r = 0.74) and reliable (all items’ ITC > 0.3; Cronbach’s α = 0.83, which would not have significantly increased after removing any of the items). Additionally, all of the experts and students made positive comments about the readability and understandability of the tool.

#### 2.3.3. Intrapersonal Communication and Self-Reflection Self-Efficacy Scale (ISR-SES)

The I-CVI of the 8 items comprising the ISR-SES ranged from 0.88–1, which means that all of them were retained for its pilot study. The ISR-SES’s temporal stability (r = 0.77) and reliability (all items’ C-ITC > 0.3; Cronbach’s α = 0.87, which would not have significantly increased if any of the items had been removed) proved to be very good after its pilot study. After analyzing the comments from the experts and the students, some minor changes to the wording of 3 items related to the concept of ‘self’ were applied in order to improve the tool’s readability and understandability. Please see Appendix A for detailed information about the comments made by experts and participants.

### 2.4. Data Analysis of the Instruments’ Final Version

The PCC-SES, PIE-SES, and ISR-SES were tested amongst the main study sample (*N* = 305) and psychometrically assessed following other authors’ procedures and recommendations [24,30,31,32]. The original version of the CC-SET was developed and tested in Spanish (see the full version of the three tools comprising the CC-SET in the Appendix A). The English translation presented in this manuscript was carried out following the forward-backward translation procedure recommended by the ‘European Organisation for Research and Treatment of Cancer’ [33] (please, consider that the English version of the CC-SET presented here has not undergone any psychometric assessment and it should not be considered as validated for use with English speaking populations). The data analysis was performed using IBM^®^ SPSS^®^ v.22. 

#### 2.4.1. Readability and Understandability

The readability and grade level of the CC-SET was objectivized using the Flesch-Kincaid tool in Microsoft Word^®^ for Mac 2016. Nine independent proficient Spanish-speakers and six non-proficient Spanish-speakers were asked about the difficulties they found when reading and completing the CC-SET. This allowed for the assessment of the tools’ understandability [24]. The amount of time needed to complete the toolkit was recorded.

#### 2.4.2. Reliability

The method to evaluate the reliability of the final version of the three tools comprising the CC-SET was identical to the one already described in the section ‘initial development and testing of the toolkit’.

#### 2.4.3. Validity

The validity of the CC-SET was assessed by exploring content, criterion and construct validity of each individual instrument. The tools’ content validity was explored by using the same method as the one described in the section ‘initial development and testing of the toolkit’. Additionally, the average CVI of each scale (S-CVI/Ave) was calculated. A S-CVI/Ave > 0.90 was considered to evidence the instruments’ content validity [30]. The tools’ criterion validity was explored by comparing the participants’ results on the PCC-SES, PIE-SES, and ISR-SES with their results on previously-validated tools that intend to measure similar constructs. The self-efficacy questionnaire developed by Axboe et al. [34] to measure clinical communication skills of healthcare professionals (SE-12) was used as the criterion of reference for comparisons with the PCC-SES. Due to the lack of specific and previously-validated tools, the New General Self-Efficacy Scale (NGSES) was used as the criterion for comparisons with the PIE-SES and ISR-SES [35]. The Pearson’s correlation coefficient (r) between the participants’ results on the tools comprising the CC-SET and the tools used for comparison was calculated. Lastly, for the evaluation of the three tools’ construct validity, exploratory factor analysis (EFA) using principal axis factoring (PAF) and known-groups analysis (KGA) was carried out.

PAF—The Bartlett’s Test of Sphericity and the Kaiser–Meyer–Olkin Measure of Sampling Adequacy tested the appropriateness of performing EFA. Then, an unlimited PAF with Varimax rotation was performed. Factors were considered a structural part of the PCC-SES, PIE-SES, and ISR-SES if they had an eigenvalue ≥1, if there was a clear break on the plot of eigenvalues, and if all items loading onto the factor did so with a factor-loading value ≥0.45 [36].KGA—The main sample (*N* = 305) was divided in groups depending on time passed since the participant last attended a training session in clinical communication. Based on this categorization, between-groups differences in the individuals’ scores for the PCC-SES, PIE-SES, and ISR-SES were expected and one-way analysis of variance (ANOVA) was used to analyze these data. Furthermore, in order to evaluate the differences between groups’ mean scores, Tukey’s honestly significant difference (HSD) post-hoc tests were also carried out.

## 3. Results

### 3.1. Description of the Main Sample

Table 1 summarizes the demographic characteristics of the main sample (*N* = 305) and the known groups that were used for the analysis of the tools’ construct validity. The sample’s mean age was 22.28 years (SD = 5.68; range = 17–52) and 80.33% were female (*n* = 245). Additionally, *n* = 296 (97.05%) participants had completed upper secondary education before entering the undergraduate nursing program and *n* = 103 (33.77%) participants were first-year students, *n* = 81 (26.56%) were third-year students and *n* = 121 (39.67%) were fourth-year students.

### 3.2. Readability and Understandability

The reading level of the PCC-SES, PIE-SES, and ISR-SES corresponds to 9th, 9th, and 6th grade respectively. The students and the independent Spanish-speakers did not report any difficulties understanding any of the three tools comprising the CC-SET. In addition, participants took between 2–5 min to complete the PCC-SES, 1–3 min for the PIE-SES, and 2–4 min for the ISR-SES. On average, participants took 7 min to complete the CC-SET.

### 3.3. Reliability

Table 2 shows the results of the reliability analysis. The Cronbach’s α for the PCC-SES, PIE-SES, and ISR-SES was 0.93, 0.87, and 0.86 respectively. Furthermore, the C-ITC ranged from 0.52–0.75 for the PCC-SES, from 0.57–0.75 for the PIE-SES, and from 0.55–0.69 for the ISR-SES. None of the tools’ Cronbach’s α would have improved if any of their items had been removed.

### 3.4. Validity

The CVI for all the items comprising the CC-SET ranged from 0.88–1 (see Table 2) and the tools’ S-CVI/Ave were 0.97 (PCC-SES), 0.98 (PIE-SES), and 0.95 (ISR-SES). Regarding criterion validity analysis, PCC-SES, PIE-SES, and ISR-SES correlated well with the tools they were compared to (r = 0.66; *p* < 0.001; r = 0.65; *p* < 0.001; r = 0.83; *p* < 0.001, respectively). The results related to the analysis of construct validity are presented below.

#### 3.4.1. PAF

According to the results of both, the Kaiser–Meyer–Olkin measure of sampling adequacy (PCC-SES = 0.92; PIE-SES = 0.82; ISR-SES = 0.81) and the Barlett’s Test of Sphericity [PCC-SES (χ^2^ = 2851.66; *p* < 0.001); PIE-SES (χ^2^ = 875.83; *p* < 0.001); ISR-SES (χ^2^ = 1016.03; *p* < 0.001)], it was appropriate to conduct EFA for the three instruments. 

Whilst the PCC-SES has three underlying factors with eigenvalues ≥1 and a clear break on the plot of eigenvalues, the PIE-SES has one and the ISR-SES has two. Table 3 and Table 4 show the results of the PAF for the PCC-SES and ISR-SES. All items comprising the three tools load onto one of the factors with a factor-loading value ≥0.45. The three factors comprising the PCC-SES account for 61.70% of the total variance found and represent different core aspects of person-centered conversation: (1) ‘starting the conversation’, (2) ‘effective person-centered communication skills’, and (3) ‘empathy and respect’. The only factor of the PIE-SES accounts for 60.19% of the total variance found and it demonstrates that this tool has a single-factor structure. Lastly, the ISR-SES has two factors that account for 64.13% of the total variance found and they represent two important dimensions of the measured construct: (1) ‘self-actualization’, and (2) ‘reflective skills’.

#### 3.4.2. KGA

The one-way ANOVA analysis showed significant differences in the mean scores between the three known-groups for the PCC-SES (F(2302) = 8.55; *p* < 0.001), PIE-SES (F(2302) = 6.04; *p* = 0.003), and ISR-SES (F(2302) = 17.47; *p* < 0.001). Table 5 presents the results for the KGA and Tukey’s HSD post-hoc tests.

## 4. Discussion

Nursing students experience difficulties when communicating to patients, patients’ relatives and other colleagues in their clinical placements [3,15,16]. It is important for nursing educators to foster the acquisition of competence in clinical communication amongst nursing students before they are faced with real human interactions during their clinical placements [17]. Standard and valid measures to comprehensively assess nursing students’ competence in clinical communication are needed [11]. This study aims to design and psychometrically evaluate a toolkit to comprehensively assess nursing students’ self-efficacy in clinical communication.

When a new measurement tool is developed, its psychometric properties must be rigorously tested in terms of reliability, validity, and usability [31]. Regarding reliability, the three tools comprising the CC-SET (PCC-SES, PIE-SES, and ISR-SES) have shown to have an excellent internal consistency and a very good temporal stability. This can be considered evidence of the CC-SET’s ability to measure accurately [31,32]. The validity of the CC-SET was assessed through evaluating the content validity, criterion validity and construct validity of the PCC-SES, PIE-SES, and ISR-SES [24,31,32]. Regarding content validity, a panel of 17 experts decided that the three tools comprising the CC-SET were appropriately designed to measure nursing students’ self-confidence to conduct person-centered conversations (PCC-SES), to communicate patients’ clinical information using the SBAR method (PIE-SES) and to communicate with oneself and self-reflect (ISR-SES) [30]. Concerning criterion validity, the PCC-SES, PIE-SES, and ISR-SES have shown to correlate strongly with the criteria of reference to which they were compared. This demonstrates that the tools comprising the CC-SET can be used to make decisions about nursing students’ self-efficacy in clinical communication [31,32]. In terms of construct validity, exploratory factor analysis has revealed that the underlying structural dimensions of the three tools comprising the CC-SET are in line with the theoretical underpinnings upon which they were developed. On the one hand, whereas the PIE-SES only had one dimension that concurs with the SBAR model [28], the PCC-SES presented three dimensions (i.e., ‘starting a conversation’, ‘effective person-centered communication skills’, and ‘empathy and respect’) that are clearly in line with the SAGE&THYME^®^ and SEGUE models [26,27]. On the other hand, the ISR-SES has demonstrated to have a two-dimension structure that clearly corresponds with the concepts of ‘self-actualization’ and ‘reflective skills’ that appear in Rogers’ and Honeycutt’s theories of intrapersonal communication [19,29]. Adding onto the CC-SET’s construct validity, known-group analysis showed that the PCC-SES, PIE-SES, and ISR-SES are capable of detecting differences between students with different levels of training. In summary, all the evidence related to the CC-SET’s validity demonstrates that this toolkit accurately measures the concept ‘self-efficacy in clinical communication’ amongst nursing students [31,32,36]. In addition to the CC-SET’s reliability and validity, our psychometric evaluation has also shown that the PCC-SES, PIE-SES, and ISR-SES are easy to complete and understand. This could facilitate its use as part of the assessment of nursing students’ competence in clinical communication.

Although the CC-SET has shown excellent psychometric properties after a rigorous initial evaluation, some limitations must be highlighted. Firstly, the participants were recruited through a convenience sampling method, which means that the study’s results can only be generalized to populations with very similar characteristics. Other studies should focus on testing the CC-SET amongst other populations in different settings. Secondly, due to organizational constraints, the CC-SET’s temporal stability was only assessed in its pilot study. It is recommended that further studies collect data in two different points in time and calculate the tools’ intraclass correlation coefficient. Thirdly, the study’s results are based on self-reported data and they could have been influenced by social desirability response bias.

## 5. Conclusions

The CC-SET has evidenced to be comprised of three tools with excellent psychometric properties to objectively and comprehensively assess nursing students’ self-efficacy in clinical communication. When used in conjunction with other tools that assess nursing students’ knowledge and skills in clinical communication, the PCC-SES, PIE-SES, and ISR-SES could help nursing educators to quickly and accurately learn what the real educational needs of their students are. It is recommended that the CC-SET be used to assess nursing students’ self-efficacy to conduct person-centered conversations, to communicate patients’ clinical information using the SBAR method and to communicate with themselves before they are exposed to real human interactions during their clinical placements. Further studies could evaluate the psychometric properties of the CC-SET in different languages and settings.

## Figures and Tables

**Table 1 ijerph-16-04534-t001:** Demographic characteristics of the main sample (*N* = 305) and known groups.

Characteristic	Main Sample (*N* = 305)	Year 1. Attended Basic Training Immediately before Testing (*N* = 103)	Year 3. Attended Basic Training 1 Year before Testing (*N* = 81)	Year 4. Attended Advanced Training 3 Months before Testing (*N* = 121)
	M ± S.D.	M ± S.D.	M ± S.D.	M ± S.D.
**Age** (years)	22.28 ± 5.68	20.18 ± 5.76	21.17 ± 4.62	24.14 ± 5.65
	*n* (%)	*n* (%)	*n* (%)	*n* (%)
**Gender**				
Female	245 (80.33)	86 (83.49)	69 (85.19)	90 (74.38)
Male	60 (19.67)	17 (16.51)	12 (14.81)	31 (25.62)
**Education Level (completed)**				
Upper Secondary Education	296 (97.05)	100 (97.08)	79 (97.53)	117 (96.69)
Degree	9 (2.95)	3 (2.92)	2 (2.47)	4 (3.31)

**Table 2 ijerph-16-04534-t002:** Psychometric statistics of item analysis for reliability and I-CVI of the CC-SET (*N* = 305).

	C-ITC ^1^	Cronbach’s α if Item Deleted	I-CVI ^2^
**Patient-Centered Communication Self-Efficacy Scale (PCC-SES)**			
**Item 1** Choose the appropriate environment for my interactions before initiating them, taking into consideration the nature of each interaction.	0.62	0.92	0.94
**Item 2** Appropriately introduce myself and greet other people taking into consideration the particularities of each interaction.	0.52	0.93	0.94
**Item 3** Explain the reason why I am interacting with other people and clarify what my time availability is before starting a conversation.	0.65	0.92	0.88
**Item 4** Ask the right questions in order to initiate a difficult conversation, taking into consideration the particularities of each situation.	0.63	0.92	1
**Item 5** Demonstrate empathy without judging the other person’s concerns.	0.65	0.92	0.94
**Item 6** Gather all the relevant information about the other person’s concerns without forcing them to share it.	0.64	0.92	0.88
**Item 7** Make an appropriate and useful use of silence during my interactions with other people.	0.69	0.92	1
**Item 8** Give other people the opportunity to express themselves without interrupting or influencing them.	0.66	0.92	0.94
**Item 9** Make an appropriate use of feedback in order to clarify what I have understood and to demonstrate that I am actively listening.	0.57	0.92	1
**Item 10** Encourage and guide people I am interacting with to identify what their support network may be without telling them myself.	0.67	0.92	0.94
**Item 11** Encourage other people to talk about what they think would help, giving them time to think about helpful solutions without suggesting my own ones.	0.72	0.92	1
**Item 12** Offer myself as a support element without being patronizing or falling into giving advice that does not meet the other person’s needs.	0.58	0.92	0.94
**Item 13** Summarize and clarify the key points of the interaction, acknowledging the other person’s feelings and concerns and including the suggested solutions and the agreed action plan.	0.75	0.92	1
**Item 14** Appropriately use all the non-verbal communication elements regardless of the situation.	0.71	0.92	1
**Item 15** Send messages that are always complete in terms of content, feelings and demand, adapting them to the needs of each person I interact with.	0.70	0.92	1
**Item 16** Be respectful with the other person regardless of their attitude and concerns.	0.52	0.92	1
**Item 17** Accept other people’s opinions, values, beliefs, concerns and peculiarities without judging them.	0.48	0.93	1
**Patient Clinical Information Exchange and Interprofessional Communication Self-Efficacy Scale (PIE-SES)**			
**Item 18** Clearly and briefly state the situation that led to establishing contact regarding the patient I may need help with or advice for.	0.67	0.84	1
**Item 19** Communicate the patient background in a comprehensive yet concise manner without forgetting any information.	0.75	0.83	1
**Item 20** Share with other healthcare professionals all the information collected during the patient assessment process in a structured, clear, and concise manner.	0.67	0.84	1
**Item 21** Recommend appropriate and potentially effective solutions that may help with the patient’s problem.	0.71	0.84	0.88
**Item 22** Justify and argue my point of view in an assertive and respectful manner when interacting with other healthcare professionals.	0.60	0.85	1
**Item 23** Listen to other professionals’ opinions and take them into account in order to find a team solution to the patients’ problem.	0.57	0.86	1
**Intrapersonal Communication and Self-Reflection Self-Efficacy Scale (ISR-SES)**			
**Item 24** Be congruent with my own ‘self’ and eliminate the differences between my ‘real-self’ and my ‘ideal-self’.	0.58	0.84	0.94
**Item 25** Accept without judging that other people’s ‘self’ may be different to mine.	0.58	0.84	0.94
**Item 26** Accept without judging that other people may see me in a different way to how I see my own ‘self’.	0.65	0.83	0.94
**Item 27** Identify and solve the differences between how I really am and how other people see me.	0.69	0.83	0.88
**Item 28** Appropriately use self-revelation as a technique to improve my interactions, adapting it to the other person, the environment and the situation.	0.55	0.85	0.88
**Item 29** Appropriately react to other people’s criticism.	0.60	0.84	1
**Item 30** Identify and recognize potential mistakes in the way I communicate.	0.66	0.83	1
**Item 31** Reflect on my interactions with other people and do what I can to improve them in the future.	0.57	0.84	1

^1^ C-ITC = corrected item-total correlation; ^2^ I-CVI = item content validity index.

**Table 3 ijerph-16-04534-t003:** Factor loadings and total variance explained from the rotated factor structure of the PCC-SES (*N* = 305).

Factor Item	1	2	3
**Starting the Conversation**			
Item 1 Consciously choose the appropriate physical setting […]	**0.77**	0.31	0.07
Item 2 Appropriately introduce myself and greet other people […]	**0.75**	0.05	0.34
Item 3 Explain the reason why I am having an interaction […]	**0.66**	0.36	0.19
Item 4 Ask the right questions in order to initiate a difficult conversation […]	**0.57**	0.40	0.10
Effective person-centered communication skills			
Item 6 Gather all the relevant information about the other person’s concerns […]	0.39	**0.50**	0.14
Item 7 Make an appropriate and useful use of silence […]	0.38	**0.60**	0.28
Item 9 Make an appropriate use of feedback […]	0.21	**0.71**	0.17
Item 10 Encourage and guide people […] to identify their support network […]	0.22	**0.78**	0.04
Item 11 Encourage other people to talk about what they think would help […]	0.36	**0.74**	0.05
Item 12 Offer myself as a support element, asking the patient if I can help […]	0.17	**0.61**	0.24
Item 13 Summarize and clarify the key points of the interaction […]	0.31	**0.76**	0.15
Item 14 Appropriately use all the non-verbal communication elements […]	0.23	**0.70**	0.29
Item 15 Send messages complete in terms of content, feelings and demand […]	0.25	**0.70**	0.22
Empathy and respect			
Item 5 Demonstrate empathy without judging the other person […]	0.41	0.36	**0.53**
Item 8 Give other people the opportunity to express themselves […]	0.34	0.35	**0.45**
Item 16 Be respectful with the other person regardless of their […]	0.32	0.21	**0.80**
Item 17 Accept other people’s opinions, values, beliefs, concerns […]	0.02	0.25	**0.84**
% of variance	18.43	29.50	13.77
Cumulative % of variance	18.43	47.93	61.70

The factor loading figures in bold indicate which factor each item loads onto.

**Table 4 ijerph-16-04534-t004:** Factor loadings and total variance explained from the rotated factor structure of the ISR-SES (*N* = 305).

Factor Item	1	2
**Self-actualization**		
Item 24 […] eliminate the differences between my ‘real-self’ and my ‘ideal-self’.	**0.80**	0.28
Item 25 Accept without judging that other people’s ‘self’ may be different […]	**0.90**	0.19
Item 26 Accept […] that other people may see me in a different way […]	**0.59**	0.37
Reflective skills		
Item 27 Identify and solve the differences between how I really am and […]	0.37	**0.71**
Item 28 Appropriately use self-revelation […]	0.39	**0.53**
Item 29 Appropriately react to other people’s criticism.	0.21	**0.75**
Item 30 Identify and recognize potential mistakes in the way I communicate.	0.18	**0.84**
Item 31 Reflect on my interactions with other people […]	0.13	**0.78**
% of variance	27.21	36.92
Cumulative % of variance	27.21	64.13

The factor loading figures in bold indicate which factor each item loads onto.

**Table 5 ijerph-16-04534-t005:** Known groups analysis and Tukey’s HSD post-hoc test for multiple comparisons.

Known-Groups	Attended Training Immediately before Testing (*N* = 103)	Attended Training 1 Year before Testing (*N* = 81)	Attended a Refresher 3 Months before Testing (*N* = 121)
Instrument	M ± SD	M ± SD	M ± SD
Known-Groups	significance	significance	significance
PCC-SES	76.47 ± 11.84	73.68 ± 10.64	79.95 ± 9.74
Attended training immediately before testing	-	0.19	0.04
Attended training 1 year before testing	0.19	-	0.001
Attended a refresher 3 months before testing	0.043	0.001	-
PIE-SES	77.78 ± 14.90	74.79 ± 11.56	80.94 ± 10.53
Attended training immediately before testing	-	0.24	0.14
Attended training 1 year before testing	0.24	-	0.002
Attended a refresher 3 months before testing	0.14	0.002	-
ISR-SES	74.24 ± 13.43	72.52 ± 10.79	81.53 ± 10.92
Attended training immediately before testing	-	0.59	0.001
Attended training 1 year before testing	0.59	-	0.01
Attended a refresher 3 months before testing	0.001	0.01	-

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
