# Peer review of "Design and Psychometric Evaluation of the ‘Clinical Communication Self-Efficacy Toolkit’"

_ijerph, 2019, doi:10.3390/ijerph16224534_

Round 1

Reviewer 1 Report

This paper outlines an initial validation of the CCSET instrument for use with nursing students.

In regards to the statement on page 3 about adding difficulty to the items, it would be helpful if the authors could explain this in some more depth so the interested reader can have confidence in the procedure employed. Likewise, please explain in more detail about the expert review process. In general, there are sections of text that could have a bit more information provided, and I've noted this in the comments in the PDF. I've offered some suggestions for the table format as well. In the discussion, I was concerned about the lack of context regarding the mention of a few models the authors note, and while references are provided, there should be at least some mention of what these models represent. Maybe this needs to be articulated in the intro section so it makes sense in the discussion? I would also ask the authors to keep in mind that the validation procedures offered in this article are acceptable, but they are relatively common and therefore referring to them as "rigorous" and "proof" that the instrument is valid might be overwhelming. There are much more rigorous validation methods out there, so there might be more emphasis on this as a being a rigorous initial validation study. The authors do mention the need for further validation as a limitation, which is good.

Author Response

Firstly, the authors would like to take this opportunity to thank the reviewers for their feedback. We think that after addressing the issues they have pointed out, our paper has significantly improved. Please, find below the authors’ response to the reviewers’ comments. Each comment has been dealt with separately. For clarity, all changes made to the original manuscript have been highlighted in yellow in the revised submitted version of the paper.

Kind regards,

The Authors.

Reviewer’s comment:In regards to the statement on page 3 about adding difficulty to the items, it would be helpful if the authors could explain this in some more depth so the interested reader can have confidence in the procedure employed.

Authors’ response:Thanks for your comment. As we explain in the manuscript, the items of the three tools were based on several communication models and guidelines. These models and guidelines include generic tasks that healthcare professionals should undertake. When we were faced with the operationalization of these tasks to convert them into scales’ items, we realise that they were to generic and people may have considered them “easy to achieve”; thus, giving themselves high scores in all items. In order to avoid these, subtle elements that added some difficulty to the task were added. For example: in item 1, instead of using “choose the environment for your interactions before initiating them”, we added the following: “Choose the appropriate environment for my interactions before initiating them, taking into consideration the nature of each interaction”, which makes the task more specific and therefore a bit more difficult. In order to make this clear in the manuscript, we have added a brief explanation. Please, see lines 129-134.

Reviewer’s comment:Likewise, please explain in more detail about the expert review process.

Authors’ response:Please see response further below.

Reviewer’s comment:In general, there are sections of text that could have a bit more information provided, and I've noted this in the comments in the PDF. I've offered some suggestions for the table format as well.

Authors’ response:Thanks for this. All the reviewer’s comments included in the pdf document have been separately responded further below.

Reviewer’s comment:In the discussion, I was concerned about the lack of context regarding the mention of a few models the authors note, and while references are provided, there should be at least some mention of what these models represent. Maybe this needs to be articulated in the intro section so it makes sense in the discussion?

Authors’ response:Thanks for highlighting this. Following the reviewer’s suggestion, several clarification statements have been added before the discussion section so that the reader familiarises himself/herself with the models/theories referenced later on in the discussion section. Please changes highlighted in lines 104-125.

Reviewer’s comment:I would also ask the authors to keep in mind that the validation procedures offered in this article are acceptable, but they are relatively common and therefore referring to them as "rigorous" and "proof" that the instrument is valid might be overwhelming. There are much more rigorous validation methods out there, so there might be more emphasis on this as a being a rigorous initial validation study. The authors do mention the need for further validation as a limitation, which is good.

Authors’ response:Thanks. Please note that we have followed the reviewer’s suggestion and have nuanced our statement about rigour. Please, see lines 328-329.

Reviewer’s 1 comments included in the pdf document.

Reviewer’s comment:"with themselves" is unclear

Authors’ response:Thanks for your comment. We have rewritten the sentence so that the expression “with themselves” is clearer to the general reader. Please see lines 37-39.

Reviewer’s comment:consequence seems like a negative term here even though that's not really the implication. Maybe use a different word?

Authors’ response:Thanks for your comment. We agree with the reviewer and we have changed the clause “as a consequence” for “in response”. Please, see line 41.

Reviewer’s comment:"clinical training" instead of "communication"?

Authors’ response:Thanks for highlighting this. After seeing the reviewer’s comment, we have realised that the sentence was unclear and have rewritten the last part following the reviewer’s suggestion. Please see line 68.

Reviewer’s comment:"have" instead of "has"

Authors’ response:Thanks for highlighting this grammatical mistake. “Has” has been replaced by “have” in line 73 as suggested by the reviewer.

Reviewer’s comment:explain this a bit more. (the reviewer refers to the element of difficulty added to the items in order to avoid the celling effect).

Authors’ response:Already responded above.

Reviewer’s comment:explain what "critically revised" means--what was it they were looking for? Also, the experts-- offer a brief mention about what credentials they have as experts (e.g., academic positions, professional licensure such as MD, etc).

Authors’ response:Thanks for this comment. A clarification of what the critical revision consisted of was added at the end of the sentence. Please see lines 136-137. Equally, following the reviewer’s suggestion, the inclusion criteria to participate as an independent expert to critically evaluate the CC-SET have been included in the manuscript. Please see lines 138-140.

Reviewer’s comment:Please note if any of the experts had differing opinions and how if there was not consensus about changes how it was resolved. If there were no issues of this kind, please indicate that.

Authors’s response:Thanks for your comment. We humbly ask the reviewer to consider that, in this study, the experts did not participate in a Focus Group to assess the CC-SET’s content validity. Instead, the Polit and Beck’s (2006) modification of the Lawshe’s (1978) statistical method to assess a tool’s content validity was used. This means that the experts were sent an email survey in which they had score each item depending on their relevance to measure the intended construct. Not all the experts scored all the items’ relevance the same, but the abovementioned method to calculate the item’s content validity index was used. This method is explained in detail in lines 142-152. Nevertheless, we appreciate that this can cause confusion for the general reader and, therefore, have modified some of the sentences and added a sentence explaining that the experts independently reviewed the tool filling up a survey via email. Please see lines 141-142.

Reviewer’s comment:It would be helpful to have a table or appendix figure outlining the suggestions, since there are so many.

Authors’ response: Thanks for this comment. As suggested, Appendix 1 includes all the comments/suggestions made by the experts and participants to improve the understandability and readability of the tools comprising the CC-SET. A reference to this appendix has been included in the manuscript. Please see lines 176-177 and 191-192. This response also applies to the following comment in which the reviewer says “previous”.

Reviewer’s comment:Is the English translation provided just for the reader? It should also be made clear that the English version was not tested and therefore not be considered validated for use with English speaking populations.

Authors’ response:Thanks for your suggestion. A clarification statement about this matter has been added onto the revised version of the manuscript. Please see lines 199-201.

Reviewer’s comment: It would be helpful if all the numbers for each factor were included, and perhaps putting the numbers on each factor in bold. That is, bold the numbers you currently have and include the other numbers in non-bold.

Author’s response:Thank you for your suggestion. Following the reviewer’s comment, all the factor loading figures have now been included in both Table 3 and Table 4, with the loading factors above 0.45 in bold as suggested. We think this helps the reader to better understand the tables.

Reviewer’s comment:"evidence", not "proof".

Authors’ response: Thanks. As suggested, the word proof has been changed for “evidence”. Please, see line 302.

Reviewer’s comment:If these models are going to be referenced, at least some info about them should be included. Simply providing the references with no context in the paragraph seems unfair to the reader.

Authors’ response:Thanks for this comment. Following the reviewer’s suggestion, the main characteristics of the models referenced have been explained in the manuscript so that the readers understand the context of part of the construct validity assessment before he/she gets to the discussion section. Please see lines 104-125.

Reviewer 2 Report

Interesting study, however for those who are not psychologist it  difficult to read and to understand for me as a medical doctor. Most of the doctors and nurses are not familial with this scale and other specialist in medial scince I think also dont know it.

It is impossible to look during the reading on the tables and scales which are published in other journals. May it is possible to include thise scales and questionares as accessory internet media. See:??? !!!

(CC-SET) was comprised of three tools: the  (PCC-SES),  Self-Efficacy Scale’ (PIE-SES), and the ‘Intrapersonal communication and Self-Reflection Self-Efficacy Scale’ (ISR-SES). The tools’ reliability, validity (content, criterion and construct) and usability were rigorously tested. The Cronbach’s alpha coefficient of the three tools comprising the CC-SET was very high and demonstrated their excellent reliability (PCC-SES=0.93; PIE-SES=0.87; ISR-SES=0.86).

Author Response

Firstly, the authors would like to take this opportunity to thank the reviewers for their feedback. We think that after addressing the issues they have pointed out, our paper has significantly improved. Please, find below the authors’ response to the reviewers’ comments. Each comment has been dealt with separately. For clarity, all changes made to the original manuscript have been highlighted in yellow in the revised submitted version of the paper.

Kind regards,

The Authors.

Reviewer’s comment:Interesting study, however for those who are not psychologist it  difficult to read and to understand for me as a medical doctor. Most of the doctors and nurses are not familial with this scale and other specialist in medial scince I think also dont know it.

Authors’ response:Thanks for this comment. After undertaking the changes suggested by another reviewer, we humbly believe that the manuscript does a good job at explaining the methods of the study to the general reader.

Reviewer’s comment:It is impossible to look during the reading on the tables and scales which are published in other journals. May it is possible to include thise scales and questionares as accessory internet media. See:??? !!!

Authors’ response:Thanks for highlighting this. We agree with the reviewer in that many validation papers do not offer the reader the opportunity to access the validated tool. Consequently, the CC-SET (in its validated and original Spanish language) has been added as supplementary material. A reference to the Appendix S2 has been added onto the manuscript. Please, see lines 196-197.

Reviewer’s comment:(CC-SET) was comprised of three tools: the  (PCC-SES),  Self-Efficacy Scale’ (PIE-SES), and the ‘Intrapersonal communication and Self-Reflection Self-Efficacy Scale’ (ISR-SES). The tools’ reliability, validity (content, criterion and construct) and usability were rigorously tested. The Cronbach’s alpha coefficient of the three tools comprising the CC-SET was very high and demonstrated their excellent reliability (PCC-SES=0.93; PIE-SES=0.87; ISR-SES=0.86).

Authors’ response:Thanks.

Reviewer 3 Report

This article tries to develop the necessary tools to see if the competence of nursing students in clinical communication is acquired and evaluated thoroughly before being exposed to real human interactions during their clinical placement.

The article is written correctly, following all the guidelines for publications of scientific articles, the objective is to design and psychometrically evaluate a set of tools to comprehensively evaluate the self-efficacy in clinical communication of nursing students.

The method is perfectly described by presenting the sociodemographic data. Before applying the test to the students, it was reviewed by a group of experts in addition to previously conducting a pilot test to see how the tool behaves before applying it to the final sample. Then, in the final sample, the statistical calculations to verify the reliability and validity of the test are correct. 

The article has gone through an ethics committee that has its approval.

The results respond to the objective set at the beginning.The tables and the data presented are correct.When a new measurement tool is developed, its psychometric properties must be rigorously tested in terms of reliability, validity and usability and this has been developed clearly and rigorously in this investigation.

The discussion is perfectly organized and Although the CC-SET has demonstrated excellent psychometric properties after a rigorous evaluation, the limitations that researchers have found in this research are highlighted. 

Conclusions respond clearly and directly to the objective that was set at the beginning.

I would like to congratulate the authors for their excellent work.

Author Response

Firstly, the authors would like to take this opportunity to thank the reviewers for their feedback. We think that after addressing the issues they have pointed out, our paper has significantly improved. Please, find below the authors’ response to the reviewers’ comments. Each comment has been dealt with separately. For clarity, all changes made to the original manuscript have been highlighted in yellow in the revised submitted version of the paper.

Kind regards,

The Authors.

Reviewer’s comments:This article tries to develop the necessary tools to see if the competence of nursing students in clinical communication is acquired and evaluated thoroughly before being exposed to real human interactions during their clinical placement. The article is written correctly, following all the guidelines for publications of scientific articles, the objective is to design and psychometrically evaluate a set of tools to comprehensively evaluate the self-efficacy in clinical communication of nursing students. The method is perfectly described by presenting the sociodemographic data. Before applying the test to the students, it was reviewed by a group of experts in addition to previously conducting a pilot test to see how the tool behaves before applying it to the final sample. Then, in the final sample, the statistical calculations to verify the reliability and validity of the test are correct.  The article has gone through an ethics committee that has its approval. The results respond to the objective set at the beginning. The tables and the data presented are correct. When a new measurement tool is developed, its psychometric properties must be rigorously tested in terms of reliability, validity and usability and this has been developed clearly and rigorously in this investigation. The discussion is perfectly organized and Although the CC-SET has demonstrated excellent psychometric properties after a rigorous evaluation, the limitations that researchers have found in this research are highlighted.  Conclusions respond clearly and directly to the objective that was set at the beginning. I would like to congratulate the authors for their excellent work.

Authors’ response:Thanks for your comments.